# Avoid Being a Shortcut Learner through Library-Based Re-Learning

## Abstract

Replay-based methods provide a promising solution to address catastrophic forgetting issue in continual learning. They try to retain previous knowledge by using a small amount of data from previous tasks stored in a fix-sized buffer. In this work, we invoke the information bottleneck principles and reveal some fundamental limitations of those methods on their effectiveness in capturing the truly important features from the prior tasks by relying on the buffer data selected according to the model's performance on known tasks. Since future tasks are not accessible during model training and buffer construction, the trained model and the buffer data tend to be biased towards making accurate predictions on the labels of known tasks. However, when new task samples are introduced along with labels, the biased model and the buffer data become less effective in differentiating samples of the old tasks from those of the new ones. Inspired by the way humans learn over time, we propose a novel relearning technique that makes use of additional past data, referred to as the *library*, to test how much information the model loses after learning the new task. We then realign the model towards those forgotten samples by training on a carefully selected small subset samples from the library for a few epochs with comparable computational cost as existing replay-based models. The experimental results on multiple real-world datasets demonstrate that the proposed relearning process can improve the performance of the state-of-the-art continual learning methods by a large margin.

## 1 Introduction

Continual learning (CL) models are capable of learning from a continuous stream of various tasks over time without forgetting previously acquired knowledge (Wang et al., 2024). Different from the relatively simple task-incremental learning (TIL), the forgetting problem arises as a central challenge for the class-incremental learning (CIL), as it seeks to balance the exploitation of new knowledge while preserving existing information, thereby enabling models to become more versatile and long-lasting learners. Replay buffer-based methods have been developed to address the forgetting problem. They integrate the concept of a "replay buffer" , which is a designated storage that contains a subset of previously learned data. In a replay buffer setting, the model $\theta$ learns from both the current task data $\mathcal{Z}_t$ and the replay buffer $C_t$ during model training on the $t$-th task. By combining the learning of the current task and the retained buffer data, the model is trained to recognize all classes without specifying which task they belong to. The replay buffer contains information of previous tasks $1, .., t-1$ that can help to mitigate catastrophic forgetting by simply including the seen classes in the $t$-th task training. However, the replay buffer is fundamentally limited given how it is constructed because the model may be biased when selecting the buffer without accessing the future task data. Although some balancing mechanics such as up-sampling can be applied during training to alleviate biases, they are insufficient to help the model regain the knowledge that has already been lost. When the buffer is selected at the end of the $t$-th task, it may have already *forgotten* about certain information of previous tasks as this information is less relevant when learning with $\mathcal{Z}_t$ and $C_t$. However, such information may become crucial when generalizing to future new tasks.

While catastrophic forgetting has been widely recognized as a major bottleneck of CL, the exact reason causing its occurrence is yet to be discovered. Most previous work attributes the inability to retain the previous task data performance as the forgetting issue. However, they do not explore the case of *wrong learning*. In the case of wrong learning, the old model learns the wrong features

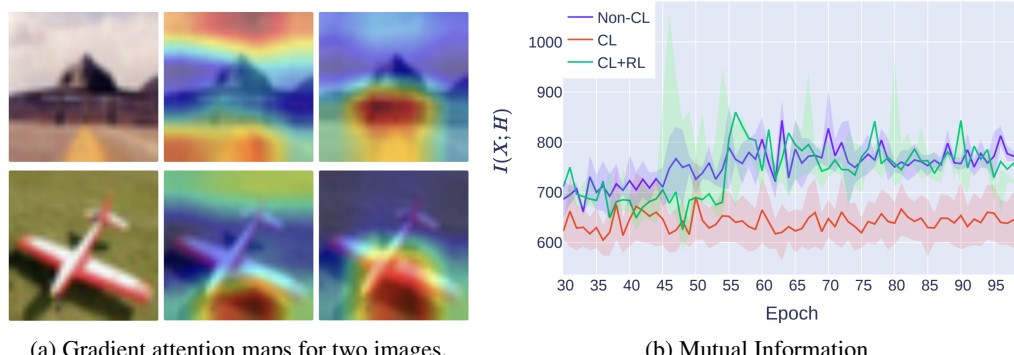

(a) Gradient attention maps for two images.  (b) Mutual Information

Figure 1: (a) An example of the shortcut issue present in the continual learning setting. (b) The effect of learning a shortcut is verified by the value of mutual information and how relearning can address it.

or reasoning to perform well only on the current task. This results in worse performance when generalizing to future tasks where the wrong features are no longer discriminative. The presence of a shortcut is shown in Figure 1a in our theoretical investigation described in Section 3. Here, the shortcut learning in the middle column is seen from training on only `Plane` and `Auto` of **Task 1** data for CIFAR-10. We call it shortcut learning because the model focuses on the background instead of the object of concern (*i.e.*, `Plane`). However, when the model is trained on both **Tasks 1&2** (`Plane`, `Auto`, `Bird`, and `Cat`) in a non-CL setting, the model then focuses on the object instead of the background (third column) as it cannot rely on the shortcut feature (sky) for classification.

In this paper, we take the first step of using the *information bottleneck principle* to explain the issue of shortcut learning that commonly exists in CL. In principle, a learning process distills the most relevant information from an input variable $X$ that is necessary for predicting an output variable $Y$, effectively compressing $X$ while retaining its predictive power for $Y$. The core idea is to find a compact representation of the input data (such as $h_t$) that maximizes the model's ability to predict the output, minimizing irrelevant or redundant information. The retained knowledge can be quantified by the mutual information. This effect is seen in the CL setting shown in Figure 1b. In **Task 1** of the split-CIFAR-10 dataset in the CL setting, the model learns to classify between only `Plane` and `Auto`. This results in the compression of the information about **Task 1** data in the hidden layer representation $I(X_1; H)$ (red plot). This compression takes place because the model can learn to classify between the classes `Plane` and `Auto` just by focusing on the sky. However, when the model is trained on combined **Task 1** and **Task 2** data (`Plane`, `Auto`, `Bird`, and `Cat`), it can no longer rely on the sky to classify between `Plane` and `Bird`. Thus, the information about the same **Task 1** data contained in the hidden layer representation $I(X_1; H)$ (blue plot) will be higher.

This key theoretical insight inspires us to draw analogy from human learning to fundamentally address the shortcut learning issue, advancing the state-of-art in CL. The connection between a machine's continual learning and real human learning is often made as continual learning mimics the human ability to accumulate and build upon past experiences (see Figure 2), thus making artificial intelligence systems more adaptable and efficient in dynamic environments. Our innovative *library* concept further strengthens this connection by drawing inspiration from how humans leverage a vast reservoir of prior knowledge to enhance learning. Just as humans access and retrieve information from past learning resources to make sense of new experiences, the library in a CL model serves a similar purpose. It acts as an expansive, accessible database from which the model can draw upon past data, significantly enriching the current learning process. This approach is reminiscent of a student revisiting notes from previous lessons or a professional drawing on past experience to solve a new problem. Although revisiting alone helps in solidifying connections within the brain, making it easier to recall information when needed, relearning reinforces the connection by using the most current understandings and allows the swap of more long-term required information. The analogy to a library-based relearning in a human learning scenario is visualized in Figure 2.

It is important to note the fundamental difference between the proposed library and the replay buffer commonly adopted by existing CL models. First, with a larger size, the library forms an accurate and unbiased approximation of the true previous task data distribution. This allows it to provide a much richer knowledge source to overcome shortcut learning as verified by our theoretical investigation.

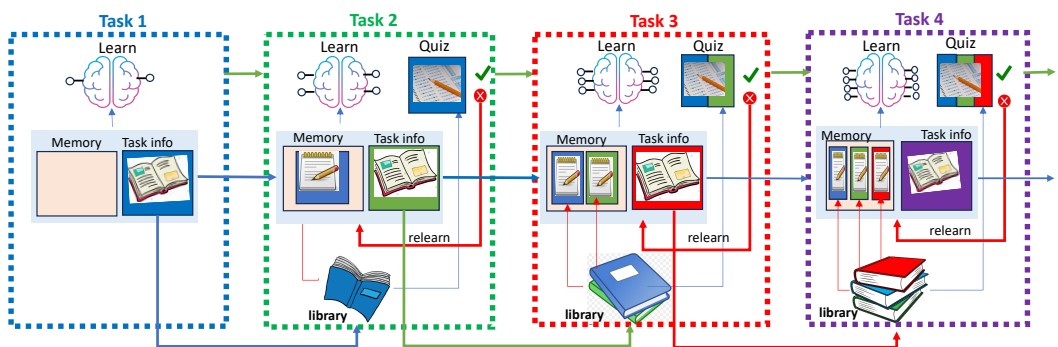

Figure 2: Continual learning with relearning from condensed past knowledge resource by humans

Second, only a small subset of data samples will be selected from the library to perform relearning if needed. By keeping the size of the selected subset to the same level as the replay buffer, the computational overhead of relearning is comparable to the replay-buffered based CL models. Last, the library also serves as a "testbed" to detect the existence of shortcut learning issue. If a model can successfully pass an active quiz uniquely designed to testing shortcut learning, relearning can be safely skipped without affecting the generalization performance on the new task. We summarize our contributions as follows:

- a novel library-based relearning framework for continual learning which better balances the reinforcement of past knowledge and acquisition of new knowledge,
- theoretical insight from an information bottleneck principle to justify the necessity of using a knowledge-rich library to overcome shortcut learning,
- an efficient and effective sample selection method for choosing informative samples from the library to perform relearning
- a uniquely design active quiz mechanism to effectively detect shortcut learning behavior that can bypass relearning,

Extensive experiments on multiple real-world datasets and comparison with the competitive CL baselines demonstrate the state-of-the-art performance of the proposed relearning framework.

## 2 RELATED WORK

A continual learner continuously updates its parameters to adapt to dynamic environments. However, during this process, there is a risk that previously acquired knowledge may be overwritten due to parameter updates, a phenomenon known as catastrophic forgetting. To address this issue, various approaches have been proposed. In this work, we categorize and discuss these methods based on whether they involve task-id prediction, to contextualize our proposed method. Additional related works are discussed in the Appendix.

**Non-task-id prediction-based CL models.** Non-task-id prediction-based methods typically use a single classification head and share all parameters across tasks. To retain knowledge from previous tasks, these methods either regularize the parameters to prevent deviation or use rehearsal examples. Regularization-based methods estimate the importance of weights with respect to previous tasks. For instance, EWC (Kirkpatrick et al., 2017) uses the Fisher information matrix to determine the strength of regularization. Orthogonal to the regularization techniques, rehearsal-based methods use small replay buffer to store a subset of examples from past tasks, periodically revisiting these examples during training. These methods generally focus on either the utilization or selection of replay examples. Particularly, beyond just using labels, they also align the behavior of the current model with the past model on the replay examples using a knowledge distillation loss to enhance performance. For example, Der++ Buzzega et al. (2020) and (Li & Hoiem, 2017) perform knowledge distillation on output logits and feature space, respectively. While those method use randomly sampling, some methods focus on the selection criteria of replay examples to maximize the performance. The sampling criterion could be choosing the most confusing examples (Chaudhry et al., 2021) or samples with least second-order influence (Sun et al., 2023).

**Task-id prediction-based CL models.** In contrast to non-task-id prediction-based methods, task-id prediction-based methods employ multiple classification heads and share a portion of parameters

across different tasks. The goal is to isolate parameters so that previously learned parameters remain fixed while learning new tasks. Specifically, we utilize the HAT model introduced by (Serra et al., 2018), which employs hard attention to selectively activate parts of the model. However, during testing, this approach relies on task-ids, which are only provided in task-incremental learning (TIL) settings. To generalize such TIL approach to class-incremental learning (CIL), several strategies have been developed to predict task IDs. For example, TPL (Lin et al., 2024) trains a separate gating network using buffer data to predict task id, but its performance depends on the availability of a large buffer. Other approaches operate independently of replay examples. For instance, MoE (Abati et al., 2020) and MOE4CL (Yu et al., 2024) train multiple autoencoders to estimate the relevance between an input and a task. $\text{HAT}_{CIL}$ (Kim et al., 2022c) and MORE (Kim et al., 2022b) leverage confidence scores from multiple classification heads to identify task-id, relying on the prediction from the most confident classification head. However, deep neural networks are known to produce overconfident predictions. In the context of continual learning, shortcut features can lead to overconfident prediction or low reconstruction score, which can undermine the effectiveness of these methods. These methods however only focus on a single task in their training objective making them susceptible to shortcut learning which affects their task-id detection performance. Our *library*-based *relearning* approach solves this problem by identifying the samples whose information is lost during the shortcut learning and *relearning* those samples.

Shortcut learning is a commonly observed behavior in deep networks, yet it is under-explored in the context of continual learning. Our work is among the few attempts to address this issue within continual learning (*i.e.,* OnPro (Wei et al., 2023)). Our approach stems from the information bottleneck perspective, and our techniques differ from the prototype learning strategy adopted by OnPro (Wei et al., 2023).

## 3 WHY RELEARN? AN INFORMATION BOTTLENECK JUSTIFICATION

**Problem Formulation.** Consider a continual learning problem where we want to learn a continual model $\theta$. At an instant or task $t \in \{1, ..., T\}$ of learning, we only have access to certain training datasets $\mathcal{Z}_t = (X_t, Y_t) = \{(x_n, y_n)\}_{n=1}^{N_t}$ and the corresponding test set $\hat{\mathcal{Z}}_t = \{(x_{\hat{n}}, y_{\hat{n}})\}_{\hat{n}=1}^{\hat{N}_t}$. The problem of *Catastrophic Forgetting* arises after the model is trained on $\mathcal{Z}_{t+1}$, as it forgets the previous data $\mathcal{Z}_t$ if we adopt no means to address it. There are two standard strategies to mitigate catastrophic forgetting. One is using a *replay buffer*, which is a small size data $C_t$ selected from $\mathcal{Z}_t$ after training the model $\theta_t$ on $\mathcal{Z}_t$. The other approach predicts the task ID from $(1, .., t)$ and then uses the corresponding classifier. An enhanced task ID prediction approach may also use $\mathcal{Z}_t + C_{t-1}$ to calibrate the task ID prediction for better generalization. In order to further improve the task ID prediction and utilize the advantages of both methods, it is beneficial to combine the task ID-based approach with replay buffers. The feature extractor is continuously trained with $\mathcal{Z}_1, ..., \mathcal{Z}_t$ and only the classification head is trained with both $\mathcal{Z}_t$ and $C_{t-1}$ during the current task.

This formulation mitigates the *Catastrophic Forgetting* issue by keeping the information of previous tasks in the feature extractor. Nevertheless, during the training of the classification head, the model is guided to minimize the training loss of both current data $\mathcal{Z}_t$ and the buffer $C_{t-1}$:

$$\theta_t^* = \arg\min_{\theta_t} \sum_{z_t^i \in \mathcal{Z}_t \cup C_{t-1}} \mathcal{L}(z_t^i, \theta_t) \tag{1}$$

The remaining issue with this way of classification is two-fold. First, similar to a standard replay buffer-based method, the model will easily overfit to the past data $C_{t-1}$ due to the emphasis on the current task (and the sizes being $|C_{t-1}| \ll |\mathcal{Z}_t|$). This causes the generalization performance to drop for those classes in $C_{t-1}$. Second, the buffer $C_{t-1}$, limited by its size, cannot capture all the information contained in the previous data $\mathcal{Z}_{t-1}$ and the model has no way of knowing how much information it misses from old data $\mathcal{Z}_{t-1}$. Although the in-task performance can be maintained in task ID prediction-based methods, this impacts the ability to differentiate between different tasks in the future and damages test performance much more. Both these issues can be categorized as *shortcut learning* issues, which can be formalized from an information theory perspective, as detailed below.

**Mutual information maximization through augmentation.** The task ID prediction part of our method helps retain the knowledge of previously learned tasks. The key idea is to continuously learn the feature extractor while maintaining learned knowledge using methods such as masking. Then we only need to differentiate the tasks when making predictions. As mentioned above, the main

advantage of doing so is that within-task performances can be well maintained. The challenges are learning holistic representations in the CIL setting and learning to accurately predict the task IDs.

To tackle the first challenge, we resort to using data augmentations during feature learning. Existing methods (Khosla et al., 2020) have connected contrastive learning with data augmentation to the maximization of the lower bound of two mutual information (MI) terms $I(X; H)$ and $I(Y; H)$. This approach is intuitive as we want the learned features to contain the most information about both the input data and target labels of the tasks. We make use of their conclusion that the lower bound of MI $I(X; H)$ can be effectively maximized through contrastive learning using augmentations: $I(X; H) \geq \max I(X; X')$, where $X'$ can be various augmentations of $X$. However, we make an important observation here that even though we maximize $I(X; H)$ within each task, the loss of information will still happen when we move on to the next task. Additionally, the second challenge also persists because augmentation only happens in-task. We contribute these issues to the shortcut learning behavior of the model. Next, we analyze this shortcut learning phenomenon in the class incremental setting using grouped features.

**Shortcut Learning from an information bottleneck perspective.** The loss of information due to shortcut learning can be explained from the perspective of *Information Bottleneck* principle. Let $Y_t$ denote the set of labels corresponding to the inputs $X_t$ for task $t$ in the CIL setting. According to the information bottleneck principle (Shwartz-Ziv & Tishby, 2017), a deep learning model compresses the information about the input $X_t$ and maximizes the information about the label $Y_t$ contained in the hidden layer representation $H_t$. The relationship between these terms can be explained by the chain rule: $I(X_t; H_t) = I(X_t; H_t | Y_t) + I(Y_t; H_t)$. Thus, although the previous method (Khosla et al., 2020) tries to maximize $I(X_t; H_t)$ and $I(Y_t; H_t)$ at the same time, what happens naturally is that the conditional MI $I(X_t; H_t | Y_t)$ will be regularized. An alternative formulation describes this process as a regularized optimization:

$$\min_{p(H_t | X_t)} [I(X_t; H_t) - \beta I(Y_t; H_t)] \tag{2}$$

where $\beta$ indicates the strength of relevant information about $Y_t$ captured in $H_t$. We then analyze how the information changes from one task to the next. Suppose for some task $t$, the model is trained on a set of two labels $y_{(1)}$ and $y_{(2)}$ where $Y_t = \{y_{(1)}, y_{(2)}\}$. Assume the dataset $X_t$ is represented by $q$ set of features $\mathcal{H} = \{h_{(1)}, h_{(2)}, ..., h_{(q)}\}$ through all the individual input $X_t$. A subset of feature $h_t \in h$ might be strongly correlated with the labels in $Y_t$. Thus, while learning to classify between labels $y_{(1)}$ and $y_{(2)}$, the model will compress the features $\mathcal{H} \backslash h_t$ and maximize the information about the features $h_t$ in the hidden layer representation $H_t$:

$$\min_{p(H_t; X_t)} [I(h \backslash h_t \in X_t; H_t) - I(h_t \in X_t; H_t)] \tag{3}$$

where $h_t$ strongly correlates with $Y_t = \{y_{(1)}, y_{(2)}\}$ and some features in $\mathcal{H} \backslash h_t$ also correlate with current task labels $Y_t = \{y_{(1)}, y_{(2)}\}$ but not as strong as $h_t$. The continual learning model $\theta_t$ performs well on the current task $t$ even though some relevant features $\mathcal{H} \backslash h_t$ about the data $X_t$ are being suppressed while learning the model representation $H_t$. Now suppose in the future task $t + 1$, we train the model on an additional set of labels $\{y_{(3)}, y_{(4)}\}$ while the previous set of labels $\{y_{(1)}, y_{(2)}\}$ is provided through the replay buffer $C_t$ such that $Y_{t+1} = \{y_{(1)}, y_{(2)}, y_{(3)}, y_{(4)}\}$. Let us consider a case where a subset of the previous set of compressed features $h_m \in \mathcal{H} \backslash h_t$ are now highly correlated with the new set of labels $Y_{t+1}$. However, since $|C_t| \ll |X_t|$, the relevant compressed features $h_m$ may not be captured by the limited buffer, i.e., $I(h_m \in C_t; H_{t+1}) < I(h_m \in X_t; H_{t+1})$. Thus, we can not recover this information loss by only including $C_t$ in future tasks.

**Relearning to counter shortcut learning.** The only way to compensate for the shortcut learning effect is to relearn 1) lost knowledge; 2) useful knowledge previously deemed unrelated to earlier tasks. In this work, we propose to let the model access previously seen data other than the buffer $C_t$. By relearning from a different source of previous task data, we reinstate the model by adding available $I(H_{t+1}; y_{(1)}, y_{(2)}, y_{(3)}, y_{(4)})$.

## 4 LIBRARY-BASED RELEARNING

The theoretical insight inspires us to design a novel method that can quiz the model on how much knowledge it has forgotten and select the optimal data points for the model to relearn. This idea

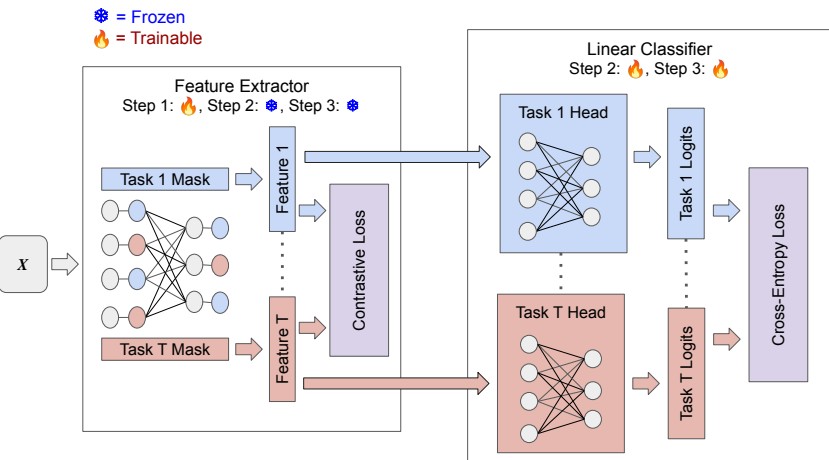

Figure 3: Relearning model architecture: STEP 1 is the feature extractor training, STEP 2 is the classifier head training, and Step 3 refers to the relearning. The input $X$ corresponds to the current data $\mathcal{Z}_t$ for step 1 and 2 where as it refers to the library buffer $C_t^L$ for STEP 3. Feature extractor is trained using the contrastive loss for step 1 and fixed for STEP 2 and STEP 3. When training task T, other task mask is kept fixed for step 1. Linear classifier is not used for step 1 and is trained for STEPS 2 & 3 using the cross-entropy loss. For STEP 2, while training task $T$, other task's classifier head is fixed. Where as for STEP 3, classifier head belonging to all the tasks are updated.

is inspired by how humans learn by storing knowledge externally other than their brain (e.g. in a library) and how they select which knowledge to relearn from the library. First, we describe the overall relearning framework and delve into detail about the relearning step.

### 4.1 OVERVIEW OF THE RELEARNING FRAMEWORK

In the proposed framework, the training of each task is divided into three steps: (1) feature extractor training,(2) classifier head training, and (3) relearning. Figure 3 visualizes the overall framework. While learning task $t$, the current data $\mathcal{Z}_t$ is augmented to $\mathcal{Z}_t'$, $|\mathcal{Z}_t'| > |\mathcal{Z}_t|$ to train a feature extractor model $f_\theta$ using a supervised contrastive loss (Khosla et al., 2020) in the feature space (STEP 1). When the feature extractor is trained for task $t$, it is frozen and a separate classifier head for task $t$ is trained using a cross-entropy loss to obtain the classification logits $\mathbf{p}_t$ (STEP 2). As the training objective only focuses on a single task, the model only learns the features important to classify current task classes. In this process, the model compresses the information needed to distinguish between the current task and future tasks. Thus, when the future task data is present to the current task classifier head, this current head will output larger prediction values for the future task data. This is how shortcut learning will hamper the task-id prediction performance. Although the *replay-buffer*-based methods can help the future task model to be calibrated with current task data, our theoretical analysis indicates that the *replay-buffer* is not enough because of lack of enough information and the smaller size of *replay-buffer*. Furthermore, just increasing *replay-buffer* size is not feasible as the computation cost of training on the *replay-buffer* will increase with the size of *replay-buffer*.

Based on our theoretical analysis, we know that storing a larger dataset from already seen past training data is necessary to quiz the model to find which information to relearn. We call this dataset the *library*. It is important to note that the *library* only serves as a potential knowledge source to overcome shortcut learning. We propose an active quiz mechanism to test the model on the *library*. Specifically, when the model is trained on the new task data, quizzing on the *library* helps us detect potential shortcut learning. Based on how well the model performs on the quiz, only a small portion of carefully selected samples will be used to perform relearning. The relearning step (STEP 3) selects a fraction of the dataset from the *library*, called the *library buffer* which is then used to relearn the classifier heads. The size of this *library buffer* remains the same as the existing replay-buffer-based approaches, avoiding additional training overhead. This enables the detection of the information lost about past data attempts to restore that information in the model. In the next section, we will describe the relearning process in detail.

## 4.2 LIBRARY-BASED RELEARNING

Building on that idea, we sample and store the seen samples in two buffer stages. First, we consider that we can store relatively large sizes of data in the library $L_t$ from the current task data $\mathcal{Z}_t$. While storing the library data is relatively cheap, it is computationally expensive to constantly train on this data. Since it is not possible to know which information will be important in the future, it is essential for the library to approximate the true distribution of the prior task data. To form an unbiased representation of this distribution, we propose to perform uniform sampling in a class-balanced manner to form the library. Since the data samples are uniformed selected into the library, not all of them are informative to address the shortcut learning issue. To this end, we proceed to select a small subset of samples from the from the *library* $L_t$ to form a *library buffer* $C_t^L$. The size of *library buffer* is chosen such that it is computationally cheaper to train on that amount of data. To select such data, we first test the model on the *library* to obtain a difficulty score for each sample in the *library*. We obtain the difficulty score $D_t^i$ for each sample based on the maximum difference between the inter-task logits $\mathbf{p}_{t'}^i$ and the maximum logit among the inter-task logits ($\mathbf{p}_t^i$):

$$D_t^i = \max \left( \mathbf{p}_{t'}^i - \max \mathbf{p}_t^i \right) \tag{4}$$

We use this score to guide our selection process for the *library buffer*. The selection is based on ranking the samples based on the difficulty score and we select the samples with a higher difficulty score. However, for the small size of the buffer (for example $m = |C_T^L| = 200$ for CIFAR10), selecting the high-difficulty score sample makes the model biased towards those samples. Thus, we switch the focus to moderately difficult samples by transforming the scores and re-ranking the samples based on the transformed scores.

Intuitively, we want a smooth function ($D_{new}^i = I(D^i)$) with a single peak to give importance to the desired difficulty level, where $D_{new}^i$ is the adjusted difficulty score. The peak of the function should be controllable. Consider the difficulty score $D^i$ is first scaled from 0 to 1 *i.e.,* $D^i \in [0, 1]$ where $D^i \to 0$ for easy samples and $D^i \to 1$ for difficult samples. Also, let the range of $D_{new}^i$ is from $-1$ to 1 *i.e.,* $D_{new}^i \in [-1, 1]$. To give priority to the most difficult samples, $(D^i, D_{new}^i)$ should be $(0, -1)$ and $(1, 1)$. Similarly to give priority to the easiest samples, $(D^i, D_{new}^i)$ should be $(0, 1)$ and $(1, -1)$. To give priority to moderately difficult samples $(D^i, D_{new}^i)$ should be $(0, 0)$, $(1, 0)$, and $(0.5, 1)$. At other points between $D^i = 0$ and $D^i = 1$, the function should be smooth. To select a func-

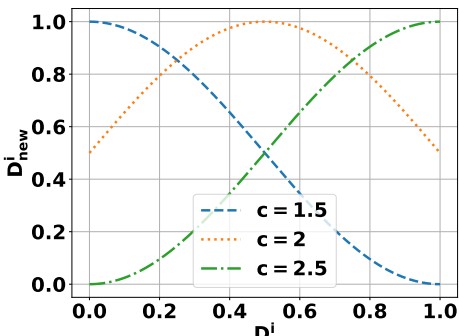

Figure 4: Visualization of the transformation function in (5). As we change the value of the parameter $c$ from 1.5 (blue) to 2.5 (green), the importance given to difficult samples increases.

tion that satisfies this property, we search on the space of smooth functions defined by a Fourier series: $I(D^i) \approx a_0/2 + a_1 \cos \left( 2\pi D^i/T_p \right) + b_1 \sin \left( 2\pi D^i/T_p \right)$ where the higher frequency components are ignored for smoothness. Here, $a_0$, $a_1$, and $b_1$ are the constant coefficients and $T_p$ is the period. We find that $D_{new}^i = I(D^i) = \sin(\pi(D^i - c))$ satisfies the above requirement of our function where $a_0 = 0$, $a_1 = -\sin \pi c$, $b_1 = \cos \pi c$, $T_p = 2$, and $c \in [1.5, 2.5]$ is the parameter that controls the location of the peak. Scale and shift are further applied on this function such that $D_{new}^i \in [0, 1]$:

$$D_{new}^i = \frac{1}{2} \left( \sin \left( \pi(D^i - c) \right) + 1 \right) \tag{5}$$

Figure 4 shows the visualization of the function in (5). The difficulty score is first scaled between 0 and 1 before applying the function. The importance given to desired difficulty scores is controlled by changing the hyperparameter $c$.

We can also consider a general case where the size of the *library* can range between the size of the *library buffer* and the size of the full set i.e, $|C_t^L| \le |L_t| \le |\mathcal{Z}_t|$. When $|C_t^L| = |L_t|$, our method is similar to the traditional replay-based method whereas when $|L_t| = |\mathcal{Z}_t|$, our method is similar to (Klasson et al., 2023) in terms of number of past samples that could be stored. For our main result, we choose $|L_t| = 0.1 \times |\mathcal{Z}_t|$ and further provide an ablation study where we increase $|L_t|$ from $|C_t^L|$ to $|\mathcal{Z}_t|$. To perform relearning, after the classifier head for the current task is trained from STEP 2,

the classifier head of all the tasks is updated by using the *library buffer*. Additionally, instead of training a new set of parameters for each task as in (Kim et al., 2022a), we update the already existing parameters of the classifier head of each task. The classifier heads are updated using only the *library buffer* $C_t^L$ which contains data from all the tasks including the current task. The computational cost involves three major parts, (1) cost of loading the *library*, (2) gradient computation, and (3) score computation. The cost of loading the *library* is $\mathcal{O}(|L_t| \times S)$ where $S$ is the size of the input image. As the *library buffer* size is small, we assume it is suitable for full-batch gradient descent whose cost is $\mathcal{O}(|C_t^L| \times |\boldsymbol{\theta_t}|)$ where $\boldsymbol{\theta_t}$ is the size of the parameter including feature extractor and the classifier head. Similarly, the score computation cost is $\mathcal{O}(|L_t| \times |\boldsymbol{\theta_t}|)$ as it requires the forward pass through both the feature extractor and the classifier head. Here we ignore the number of tasks and number of transformations as they are small compared to $|L_t|$ and $|C_t^L|$. Thus, the computational cost linearly increases with $|C_t^L|$ and $|L_t|$.

To make the task-id prediction more robust, we regularize the value of prediction logits such that the difference between the maximum prediction of within task logits and the maximum prediction among other task logits is maximized. Thus, the overall relearning loss is the sum of cross-entropy loss and the hinge regularizer.

$$\mathcal{L}_t^i = \mathcal{L}_{CE}^i + \max\left(0, \gamma + \max \mathbf{p}_{t'}^i - \max \mathbf{p}_t^i\right) \qquad (6)$$

where $t$ is the task-id for sample $i$, $\mathbf{p}_{t'}^i$ is the subvector containing the logits for classes belonging to tasks other than $t$ and $\mathbf{p}_t^i$ is the subvector containing the logits for classes belonging to task $t$. $\gamma$ is the hyperparameter which controls the strength of the regularizer.

We also develop a method that helps us decide when to perform relearning on the current task. This could be determined by quizzing the current model on the *library*. The samples with the incorrect task-id prediction will have the difficulty score greater or equal to zero. We use the count of samples in the library with $D_t^i \geq 0$ as a metric to evaluate the model and use it to guide the decision to whether to perform relearning. Specifically, we measure the count of non-negative score samples for each task: $\{|\boldsymbol{D}_j \geq 0|\}_{j=1}^t$ to get the worst performing task count $D_{max} = \max\{|\boldsymbol{D}_j \geq 0|\}_{j=1}^t$. We then set a threshold $\lambda$ such that if $D_{max} \geq \lambda$, we perform relearning. Appendix B provides the detailed algorithm of our method.

**Relearning for non-task-ID-based replay methods.**   Relearning is a general approach that can be applied to any standard replay buffer-based methods. Our theoretical analysis also holds for the replay-only setting, as the buffer will suffer from shortcut learning issues in the same way. Relearning makes up for the inability to regain lost knowledge about previously unknown useful features that can not be stored in the buffer. Empirically, we also show that relearning improves the performance of non-task-ID-based replay methods greatly in Table 2. More details about the relearning analysis for non-task-ID-based replay methods will be presented in the Appendix.

**Empirical evidence for relearning.**   Figure 1 provides empirical evidence that supports the information bottleneck based theoretical analysis as presented in Section 3. The first column in Figure 1 (a) is the original image, the second column is the attention map obtained from a continual learning model (trained on `Plane` and `Auto` from Cifar10), and the third column is the attention map obtained from the non-continual learning model (trained on `Plane`, `Auto`, `Bird`, and `Cat`). When the model is trained with classes that are closer to each other (such as `Plane` and `Bird`) the model is unable to rely on learning the shortcut feature `sky` (third column) compared to when the model is only trained on **Task** 1 data with classes `Plane` and `Auto` in the continual learning setting (second column). Figure 1 (b) shows the continual learning setting, where only training on `Plane` and `Auto` learns shortcuts that cause the mutual information value to remain lower (red plot) compared to the non-continual learning setting which has a higher value of mutual information for the same data (blue plot). The green plot shows the effect of relearning on the mutual information value of task 1 data while training the model on **Task** 2 data. The relearning happens at epoch 55 of **Task** 2 which raises the mutual information value of **Task** 1 data, demonstrating its importance.

## 5   EXPERIMENTS

**Experimental settings and baselines.**   We follow the experiment setting of recent works and evaluate in a class-incremental setting which is created by $N_C/N_T$ splitting where the dataset is split into $N_T$ tasks and each task contains $N_C$ number of classes. For each task, the feature extractor is first trained for 700 epochs using supervised contrastive loss. We follow a similar augmentation

Table 1: Comparison results with the competitive baselines

| Category | Baseline | CIFAR10-5T | CIFAR100-10T | TinyImg-10T | CF100-20T | TinyImg-5T |
|---|---|---|---|---|---|---|
| **Optimization** | OWM | $51.8_{\pm0.05}$ | $28.9_{\pm0.60}$ | $8.6_{\pm0.42}$ | $24.1_{\pm0.26}$ | $10.0_{\pm0.55}$ |
| **Regularization** | MUC | $52.9_{\pm1.03}$ | $30.4_{\pm1.18}$ | $17.4_{\pm0.17}$ | $14.2_{\pm0.3}$ | $33.6_{\pm0.19}$ |
| | PASS | $47.3_{\pm0.98}$ | $33.0_{\pm0.58}$ | $19.1_{\pm0.46}$ | $25.0_{\pm0.69}$ | $28.4_{\pm0.51}$ |
| | LwF | $54.7_{\pm1.18}$ | $45.3_{\pm0.75}$ | $24.3_{\pm0.26}$ | $44.3_{\pm0.46}$ | $32.2_{\pm0.50}$ |
| | iCaRL | $63.4_{\pm1.11}$ | $51.4_{\pm0.99}$ | $28.3_{\pm0.18}$ | $47.8_{\pm0.48}$ | $37.0_{\pm0.41}$ |
| | DER++ | $66.0_{\pm1.20}$ | $53.7_{\pm1.30}$ | $30.5_{\pm0.47}$ | $46.6_{\pm1.44}$ | $35.8_{\pm0.77}$ |
| **Replay** | Mnemonics | $64.1_{\pm1.47}$ | $51.0_{\pm0.34}$ | $28.5_{\pm0.72}$ | $47.6_{\pm0.74}$ | $37.1_{\pm0.46}$ |
| | BiC | $61.4_{\pm1.74}$ | $52.9_{\pm0.64}$ | $33.8_{\pm0.40}$ | $48.9_{\pm0.54}$ | $41.7_{\pm0.74}$ |
| **Task-id** | HAT | $62.7_{\pm1.45}$ | $41.1_{\pm0.93}$ | $29.8_{\pm0.65}$ | $25.6_{\pm0.51}$ | $38.5_{\pm1.85}$ |
| | HyperNet | $53.4_{\pm2.19}$ | $30.2_{\pm1.54}$ | $5.3_{\pm0.50}$ | $18.7_{\pm1.10}$ | $7.9_{\pm0.69}$ |
| | Sup | $62.4_{\pm1.45}$ | $44.6_{\pm0.44}$ | $36.5_{\pm0.36}$ | $34.7_{\pm0.30}$ | $41.8_{\pm1.50}$ |
| | CLOM | $87.35_{\pm0.72}$ | $65.37_{\pm0.54}$ | $47.28_{\pm0.52}$ | $57.97_{\pm0.23}$ | $50.73_{\pm0.68}$ |
| | CLOM+c | $88.34_{\pm0.44}$ | $66.06_{\pm0.02}$ | $47.59_{\pm0.53}$ | $58.89_{\pm0.20}$ | $51.02_{\pm0.66}$ |
| | TPL | $78.4_{\pm0.78}$ | $62.20_{\pm0.52}$ | $42.90_{\pm0.45}$ | $55.8_{\pm0.57}$ | $48.2_{\pm0.64}$ |
| | Relearn (ours) | $\mathbf{88.79}_{\pm0.62}$ | $\mathbf{68.80}_{\pm0.51}$ | $\mathbf{49.00}_{\pm0.28}$ | $\mathbf{61.88}_{\pm0.19}$ | $\mathbf{51.22}_{\pm0.55}$ |

technique as (Kim et al., 2022a;c). Then the classifier head training is trained for 100 epochs using cross-entropy loss. For *relearning*, the classifier head is then loaded for all tasks and trained for another 100 epochs using the selected *library buffer*. This *relearning* phase only starts from task 2.

We evaluate our method using five sequential versions of three real-world datasets: *CIFAR10-5T*, *CIFAR100-10T*, *CIFAR100-20T*, *TinyImagenet-5T*, and *TinyImagenet-10T*. The *CIFAR10-5T* dataset is created by dividing the CIFAR10 dataset into 5 tasks where each task contains 2 classes. The *CIFAR100-10T*/*CIFAR100-20T* datasets contain 10/20 tasks with 10/5 classes for each task. Similarly, the *TinyImagenet-5T*/*TinyImagenet-10T* dataset consists of 5/10 tasks and 40/20 classes each. We set the total *library* size $|L_T|$ to 5000, total library buffer size $|C_T^L|$ to 200 for CIFAR10 and 2000 for CIFAR100 and TinyImagenet datasets.

The baselines we compare can be categorized into *orthogonal projection/optimization based* methods: OWM (Zeng et al., 2019); *regularization based* methods: MUC (Liu et al., 2020b), PASS (Zhu et al., 2021), LwF (Li & Hoiem, 2017), iCaRL (Rebuffi et al., 2017), and DER++ (Buzzega et al., 2020)); *replay based* methods: (Mnemonics (Liu et al., 2020a), and Bic (Wu et al., 2019)); *task-id/parameter isolation based* methods: (HAT (Serra et al., 2018), HyperNet (Von Oswald et al., 2019), Sup (Wortsman et al., 2020), CLOM (Kim et al., 2022a), and TPL (Lin et al., 2024)). More details of the baselines are given in the Appendix. Our method of *relearning* is orthogonal to any existing CL methods and can be used to improve upon these baselines. With CLOM (Kim et al., 2022a) using a task ID prediction-based approach and being the strongest baseline, we select this baseline to perform *relearning* for our main results. Combinations with other baselines are also investigated.

### 5.1 COMPARISON RESULTS

Table 1 shows the class incremental learning result for the sequential combination of three datasets CIFAR10, CIFAR100, and TinyImagenet. The baseline results are noted as reported on the results of the works in Kim et al. (2022a) and Lin et al. (2024). We apply *relearning* on the strongest baseline (Kim et al., 2022a) and are able to further improve the performance. The most competitive baseline (Kim et al., 2022a) also uses a separately held out-validation set for the replay buffer $C_t$ to train parameters $\mathbf{W}$ and $\mathbf{b}$ to transform (scale and shift) the prediction logits of each task's classifier such that the logits of all tasks are calibrated with each other. This result is indicated in the table as CLOM+c. The use of a buffer like that of (Kim et al., 2022a) is not fair because it requires additional validation datasets unseen during the training. However, *relearning* uses already-seen samples during training and is still able to outperform the baseline.

The baselines cannot match *relearning* performance because without the *library* it is not possible to find the information that was missed by the shortcut learning. Even when the baseline uses the same size of *replay-buffer* (CLOM+c, DER++, BiC) as that of *library buffer*, the *replay-buffer* is not enough to capture the information lost during shortcut learning. With the help of *library*, we can know what data to select in the *library buffer* such that the information lost during the shortcut learning can be captured. One interesting pattern is when the number of tasks increases, the performance gap between the baseline and our method increases. This is because as new tasks appear, shortcut learning affects multiple tasks resulting in poor task-id detection performance.

Table 2: Relearning with best baselines from different categories

| Category | Baseline | CIFAR10-5T | CIFAR100-10T | TinyImg-10T |
|---|---|---|---|---|
| **Regularization** | Der++ ($m = 500$) | $71.9_{\pm 0.71}$ | $37.25_{\pm 2.06}$ | $18.77_{\pm 1.02}$ |
| | Der+++RL ($m = 500$) | $\mathbf{74.24}_{\pm 0.31}$ | $\mathbf{40.3}_{\pm 1.09}$ | $\mathbf{20.89}_{\pm 0.72}$ |
| **Replay** | BiC ($m = 500$) | $74.13_{\pm 1.00}$ | $34.26_{\pm 1.85}$ | $14.24_{\pm 1.18}$ |
| | BiC+RL ($m = 500$) | $\mathbf{79.37}_{\pm 1.61}$ | $\mathbf{43.25}_{\pm 0.84}$ | $\mathbf{24.15}_{\pm 0.99}$ |
| **Task-id** | CLOM | $87.35_{\pm 0.72}$ | $65.37_{\pm 0.54}$ | $47.28_{\pm 0.52}$ |
| | CLOM+c | $88.34_{\pm 0.44}$ | $66.06_{\pm 0.03}$ | $47.59_{\pm 0.53}$ |
| | CLOM+RL | $\mathbf{88.79}_{\pm 0.62}$ | $\mathbf{68.80}_{\pm 0.51}$ | $\mathbf{49.00}_{\pm 0.28}$ |

In Table 2, the best baseline from *regularization based*, *replay based* and *task-id based* category is selected to perform *relearning*. Here the original baseline is compared with the corresponding *relearning* (+RL) version. For Der++ and BiC, we use fixed *replay buffer* and *library buffer* size $m = 500$ for all datasets. For CLOM, we use the same setting as that of the main result in Table 1. As can be seen, our method is orthogonal to the CL baselines and can be used to improve them.

## 5.2 ABLATION STUDY

Figure 5 shows the change of final accuracy with the difficulty of *library buffer* chosen from the *library* for the CIFAR100-10T dataset. The *library buffer* of fixed size (2000) is chosen from the different sizes of *library* using equation 5. The *library buffer* is then used for *relearning* on the CLOM baseline. The result suggests it is not necessary for the size of the library to be very large. As the difficulty of samples chosen in the *library buffer* increases, the use of the larger library size starts to become less beneficial. This is because, for larger *library* size, there may be a higher chance to select outliers. When such outliers are further selected in the *library buffer*, it negatively affects the *relearning*. Thus, having relatively small and representative samples in the *library* is more beneficial and further

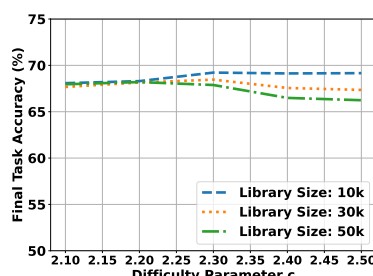

Figure 5: Impact of library size on CIFAR100-10T dataset for different difficulty level.

allows the selection of relatively more difficult samples into the *library buffer*. We find that having a *library* size from 5000 to 10000 gives the best balance.

Table 3 demonstrates our active quiz mechanism to control the time to perform *relearning* for CIFAR10-5T. At the end of each task, the *library* is queried to get the worst non-negative score count $D_{max}$ among the seen tasks. The *relearning* for the current task is only performed when $D_{max} \geq \lambda$. For this experiment, $\lambda$ is set to 100. *RL* corresponds to the performance when *relearning* is performed based on the criteria. *All RL* always performs *relearning* and *No RL*

Table 3: Actively quizzing the model to determine when to perform *relearning* for CIFAR10-5T.

| Task | $D_{max}$ | $D_{max} \geq \lambda$ | RL | All RL | No RL |
|---|---|---|---|---|---|
| 1 | - | - | 99.45 | 99.45 | 99.45 |
| 2 | 19 | No | 93.85 | 94.7 | 93.85 |
| 3 | 89 | No | 89.12 | 89.75 | 89.12 |
| 4 | 112 | Yes | 88.8 | 89.06 | 88.45 |
| 5 | 78 | No | 88.03 | 88.67 | 87.67 |
| Mean | - | - | 91.85 | **92.33** | 91.71 |

never performs *relearning* regardless of the value of $D_{max}$. Always performing *relearning* is better than selective *relearning* however, the selective *relearning* significantly reduces the computation cost as fewer task needs to be relearned. Additional experiment results are included in the Appendix C.

## 6 CONCLUSION

We perform a theoretical investigation on the shortcut learning issue that commonly exists in modern CL models from the lens of the information bottleneck principle. We address this issue by proposing a novel relearning framework, which relies on a knowledge-rich library that forms an accurate unbiased approximation of the entire data distribution of the previous tasks. The library provides a "testbed" to identify the occurrence of shortcut learning through our uniquely designed active quizzes. Relearning is invoked only if the model fails the test and a small subset of most informative data samples will be selected from the library to perform relearning so that the computational overhead is comparable to relay-based CL model training. We show that the relearning complements state-of-the art CL model and can be used to boost their performance by a large margin.

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

# Appendix

## Table of Contents

**Organization of the Appendix.** The appendix is organized as follows. Appendix A describes the additional related works in detail. Appendix B provides the algorithm of our method. In Appendix C, we provide the experiment details and results. In Appendix D, we discuss the social impact and limitations of the work.

# A  ADDITIONAL RELATED WORKS

## A.1  CONTINUAL LEARNING

Catastrophic forgetting is one of the major challenges of continual learning (Wang et al., 2024). It refers to the situation where a machine learning model dramatically forgets the knowledge for prior tasks when being trained on new data for a new task. The current proposed solutions target different steps during the training process, including replay-based (Chaudhry et al., 2019; Lopez-Paz & Ranzato, 2017; Chaudhry et al., 2018; Sun et al., 2022; 2023), architecture-based (Jin & Kim, 2022; Gurbuz & Dovrolis, 2022; Douillard et al., 2022; Wang et al., 2023), representation-based (Pham et al., 2021; Gallardo et al., 2021; Wang et al., 2022b; Zhang et al., 2023), optimization-based (Liu & Liu, 2022; Wang et al., 2022a; 2021), and regularization-based (Rebuffi et al., 2017; Benzing, 2022; Park et al., 2019; Lin et al., 2022; Jung et al., 2020). Replay-based continual learning methods store a small amount of previous data to be included in the training process. Architecture-based continual learning methods modify the neural network structure, such as increasing network size, to accommodate new information as well as retain old knowledge. Representation-based methods focus on learning a meaningful and general representation through regularization, feature distillation, or generative replay. Optimization and regularization-based methods design loss functions to penalize the neural network weight adjustment that would causes the forgetting of prior knowledge, in order to achieve a balance between learning new knowledge and retraining old knowledge.

A recent work based on influence function (Sun et al., 2023) argues that the buffer selection suffers from incidental bias. This bias occurs when we have the same or fewer samples in the pool (buffer) as we go towards the subsequent tasks. This suggests a need to have a wide variety of slightly larger data in our collection from which we can choose a smaller buffer which is then used to train a model. This motivates and justifies the use of the library in our work. As the stored library is only an order magnitude larger than the buffer size, the memory requirement is compared to the methods that store auxiliary information, augment the dataset, or train a generative model to replay the previous dataset. Furthermore, the computational cost is the same as that of the past methods because we use the same buffer size as the past methods to train the model.

## A.2  INFORMATION THEORY

The information theory in deep learning was first introduced by (Shwartz-Ziv & Tishby, 2017). They show how the DNNs compress the information about the input and maximize the information about the labels contained in the representation as the training progresses for a larger number of iterations. (Saxe et al., 2018) shows that compression only occurs for a limited number of activation functions such as tanh and sigmoid activation functions. They also show how information compression can still take place for irrelevant datasets. (Kawaguchi et al., 2023) introduce the conditional mutual information and give improved generalization bound using the conditional mutual information. In our work, we use mutual information values to explain the effect of shortcut learning in the continual learning setting which worsens the issue of catastrophic forgetting.

# B  ALGORITHM

The *relearning* approach is also described in algorithm 1. To learn a task $t$, first the feature extractor $f_{\boldsymbol{\theta}_t}$ parameterized by $\boldsymbol{\theta}_t$ is trained using the current task data $\mathcal{Z}_t$. The current data is augmented and supervised contrastive loss is used in the feature space. In STEP 2, a linear classifier head $g_{\boldsymbol{\theta}_t^g}$ parameterized by $\boldsymbol{\theta}_t^g$ for task $t$ is trained using cross-entropy loss. *Relearning* takes place in STEP 3 where the classifier heads from task 1 to $t$ are updated using the *library buffer*. *Library* is selected randomly from the current dataset to make a selection unbiased and diverse. *Library buffer* is selected from *library* using equation 5. During inference, the task-id is predicted such that the classifier head of the task with the highest logit is selected to make the prediction.

---

**Algorithm 1** Continual Learning with Relearning (RL)

### Step 1: Feature Extractor

**Input**: Dataset $\mathcal{Z}_t$
**Output**: Learned Feature Extractor $f_{\boldsymbol{\theta}_t}$

1: If $t = 1$: Initialize model $\boldsymbol{\theta}_t \leftarrow \boldsymbol{\theta}_t^0$
2: Augment $\mathcal{Z}_t$ to $\mathcal{Z}_t'$
3: Train $\boldsymbol{\theta}_t$ on $\mathcal{Z}_t'$ using supervised contrastive loss in feature space
4: **return** $f_{\boldsymbol{\theta}_t}$

### Step 2: Linear Classifier

**Input**: Dataset $\mathcal{Z}_t$, Feature extractor $f_{\boldsymbol{\theta}_t}$
**Output**: Learned Classifier Head $g_{\boldsymbol{\theta}_t^g}$

1: Initialize classifier head $\boldsymbol{\theta}_t^g \leftarrow \boldsymbol{\theta}_t^{g^0}$
2: Train $\boldsymbol{\theta}_t^g$ on $\mathcal{Z}_t$ using cross-entropy loss
3: **return** $g_{\boldsymbol{\theta}_t^g}$

### Step 3: Relearning

**Input**: Dataset $\mathcal{Z}_t$, feature extractor $f_{\boldsymbol{\theta}_t}$, classifier heads $\{g_{\boldsymbol{\theta}_1^g}, ..., g_{\boldsymbol{\theta}_t^g}\}$
**Output**: Relearned classifier heads $\{g_{\boldsymbol{\theta}_1^g}, ..., g_{\boldsymbol{\theta}_t^g}\}$

1: If $t = 1$: Select *library* $L_1 \leftarrow \mathcal{Z}_1$
2: If $t = 1$: Select *library buffer* $C_1^L \leftarrow L_1$
3: If $t > 1$: Select and append *library* $L_t = L_{t-1} \cup \{L_t \leftarrow \mathcal{Z}_t\}$
4: If $t > 1$: Select and append *library buffer* $C_t^L = C_{t-1}^L \cup \{C_t^L \leftarrow L_t\}$
5: Train $\{g_{\boldsymbol{\theta}_1^g}, ..., g_{\boldsymbol{\theta}_t^g}\}$ on $C_t^L$ using relearning loss
6: **return** $\{g_{\boldsymbol{\theta}_1^g}, ..., g_{\boldsymbol{\theta}_t^g}\}, f_{\boldsymbol{\theta}_t}$

### Inference

**Input**: Dataset $\mathcal{Z}^i$, feature extractor $f_{\boldsymbol{\theta}_t}$, classifier heads $\{g_{\boldsymbol{\theta}_1^g}, ..., g_{\boldsymbol{\theta}_t^g}\}$
**Output**: Predicted target $\hat{y}^i$

1: Get prediction logit $\{\mathbf{p}_t^i\}_{t=1}^T = \{g_{\boldsymbol{\theta}_t}\left(f_{\boldsymbol{\theta}}(\mathcal{Z}^i)\right)\}_{t=1}^T$
2: Get task id $t_{id} = \arg\max_t\{\max \mathbf{p}_t^i\}_{t=1}^T$
3: $\hat{y}^i = \arg\max \mathbf{p}_{t_{id}}^i + |\mathbf{p}_t^i| \times (t_{id} - 1)$, where $|\mathbf{p}_t^i|$ is the number of class per task
4: **return** $\hat{y}^i$

---

## C   ADDITIONAL DETAILS OF EXPERIMENTS AND RESULTS

This section includes a description of the baseline used in the main paper C.1 followed by additional ablations C.2, relearning analysis C.3, model calibration C.4 and computation resource C.5.

### C.1   DESCRIPTION OF BASELINES

In the main result shown in Table 1, we compare with the following baselines: OWM (Zeng et al., 2019) is an *orthogonal projection*-based method that changes the parameter in a direction that is orthogonal to the previous input space. MUC (Liu et al., 2020b) is a *regularization-based* method that uses an ensemble of multiple classifiers. PASS (Zhu et al., 2021) is a regularization-based replay-free method that uses regularization in the feature space. LwF (Li & Hoiem, 2017) is a regularization-based method that uses the prediction from the previous output head. iCaRL (Rebuffi et al., 2017) is also a regularization-based method that uses knowledge distillation on both old and new task data. Mnemonics (Liu et al., 2020a) is a replay-based approach that modifies training samples to make them more representative. Bic (Wu et al., 2019) is also a replay-based method that uses a bias-correction layer along with a validation dataset. DER++(Buzzega et al., 2020) is both a replay and regularization-based method that tries to solve forgetting by preventing the model from deviating too much from the past task logits. HAT (Serra et al., 2018) is a parameter isolation-based method that learns a mask to protect important parameter for each task. HyperNet (Von Oswald et al., 2019) is a parameter isolation-based method that uses task-specific parameters based on task-id and uses entropy to predict task-id. Sup (Wortsman et al., 2020) is also a parameter isolation-based approach that learns a mask to isolate parameters for each task. CLOM (Kim et al., 2022a) is also

## C.2 ADDITIONAL ABLATION

Similar to Figure 5 in the main paper, Figure 6 shows the change of final accuracy with the difficulty of samples in the *library buffer* for TinyImagenet-10T dataset. The pattern is consistent with the CIFAR100-10T dataset where a relatively small *library* size is more beneficial. As the *library* size increases, selecting more difficult samples hurt the performance.

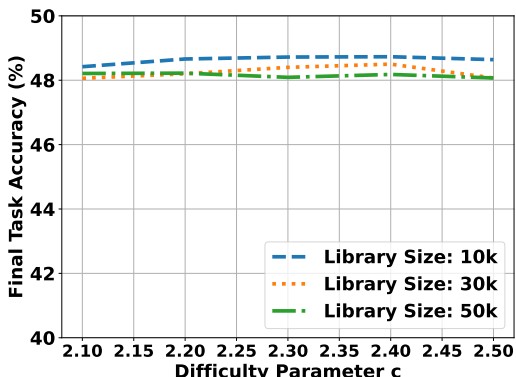

Figure 6: Impact of library size on TinyimageNet-10T dataset for different difficulty level.

Figure 7 shows the change of final performance with the parameter of the hinge regularizer $\gamma$ in (6). $\gamma = 0$ refers to without using the hinge regularizer and only using the cross-entropy loss. In our main result, we use $\gamma = 5$ for TinyImagenet, and CIFAR100 and $\gamma = 1$ for CIFAR10. Using a hinge regularizer helps the task-id prediction by maximizing the gap between the maximum logits between within task and other tasks.

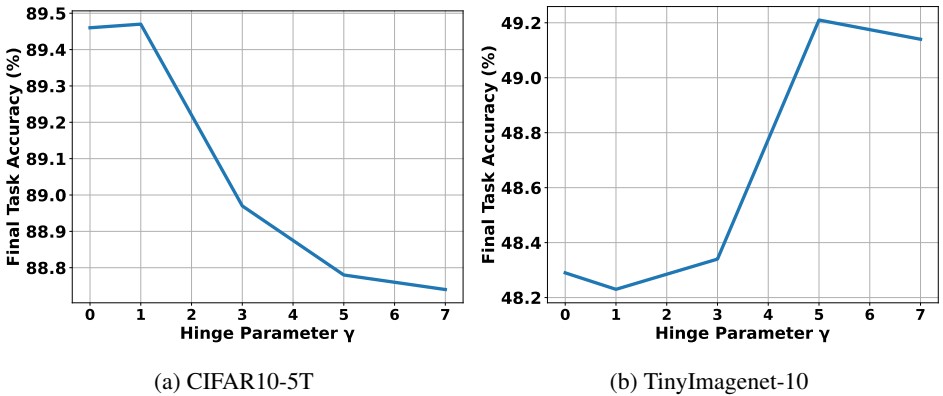

(a) CIFAR10-5T        (b) TinyImagenet-10

Figure 7: The change of final accuracy for different sizes value of hinge parameter $\gamma$.

## C.3 RELEARNING ANALYSIS

Figure 8 shows the test performance and library performance before and after relearning. The relearning phase takes place at the end of each task (task > 1) and is highlighted in the green background. We chose ER (Chaudhry et al., 2019) as a baseline for the demonstration. The library accuracy decreases as the model learns the current task while the current accuracy increases steadily. The library accuracy reflects the deteriorated performance in earlier classes and may impact the test accuracy as seen in several tasks. However, with our proposed relearning strategy, the test

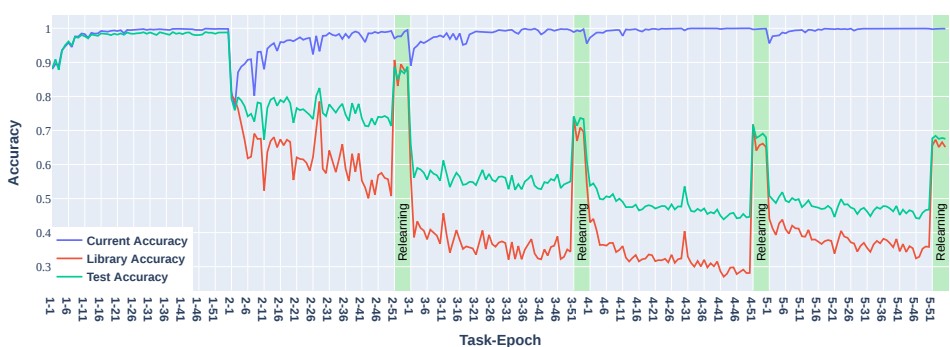

Figure 8: Effect of Relearning on the Test Accuracy

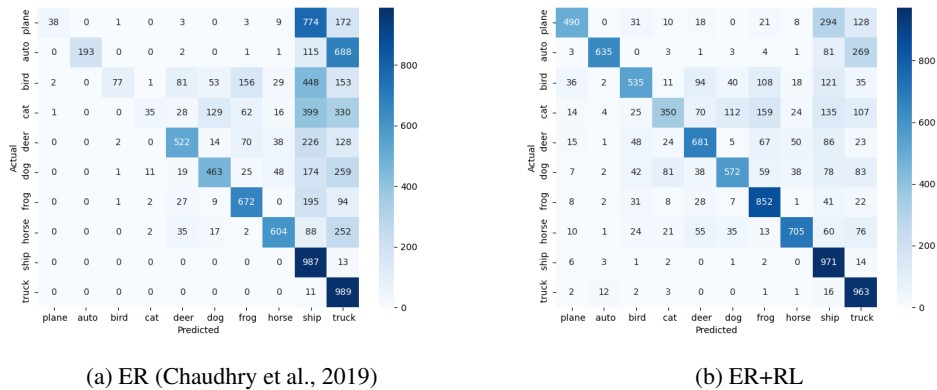

(a) ER (Chaudhry et al., 2019)

(b) ER+RL

Figure 9: Non-task-id Prediction-based Method Confusion matrix on CIFAR10-5T

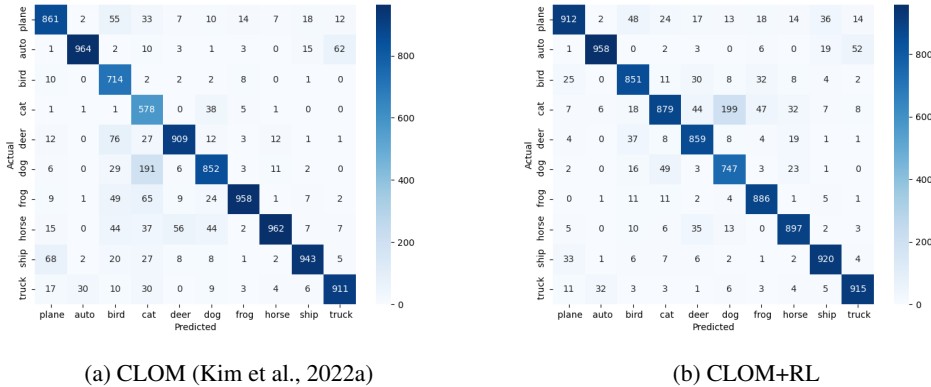

(a) CLOM (Kim et al., 2022a)

(b) CLOM+RL

Figure 10: Task-id Prediction-based Method Confusion matrix on CIFAR10-5T

performance increases simultaneously with the library accuracy by a large margin for each task. Figure 9 shows the confusion matrix for the final task of the ER (Chaudhry et al., 2019) baseline. Without relearning, the continual learning model is biased towards the current task. Relearning alleviates the bias toward the current task. Furthermore, the confusion shows inter-class conflicts, for instance, between "cat-dog" and "plane-ship-bird". For non-task-id prediction-based methods, we follow the training setting of Buzzega et al. (2020) and the *library* contains only the previous task samples.

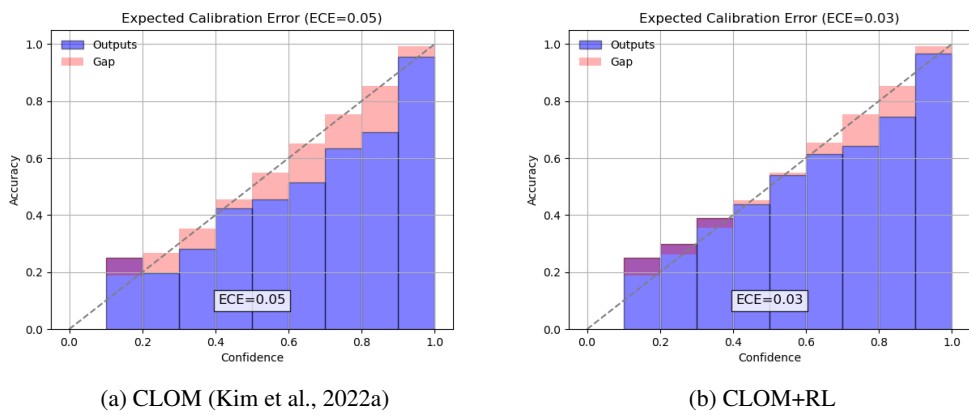

(a) CLOM (Kim et al., 2022a)   (b) CLOM+RL

Figure 11: Model Calibration on CIFAR10-5T

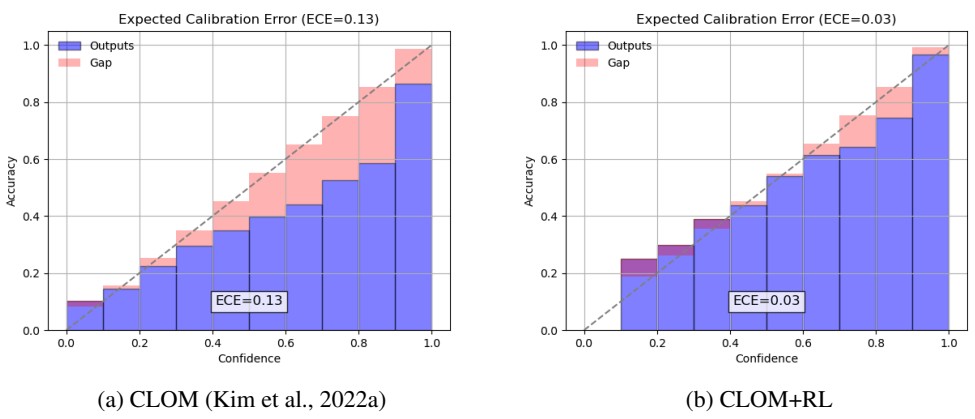

(a) CLOM (Kim et al., 2022a)   (b) CLOM+RL

Figure 12: Model Calibration on CIFAR100-10T

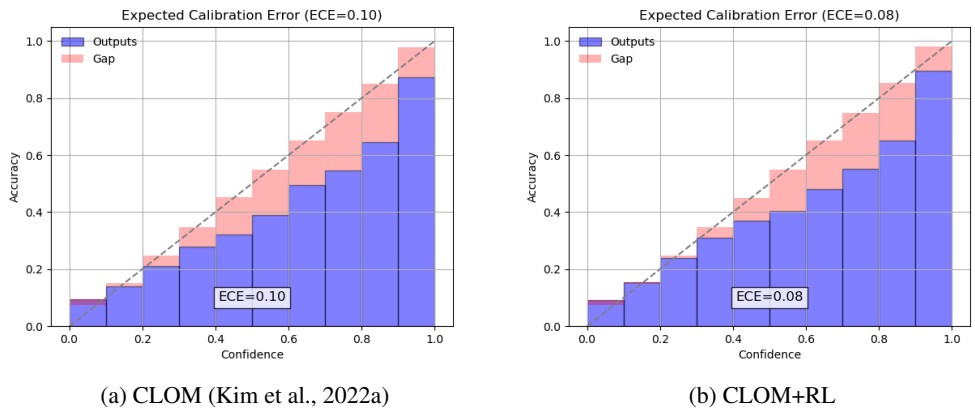

(a) CLOM (Kim et al., 2022a)   (b) CLOM+RL

Figure 13: Model Calibration on TinyImagenet-10T

For the task-id prediction-based method in Figure 10, the confusion matrix shows bias towards certain tasks that learn shortcut features like tasks 2 and 3 for CIFAR10-5T. The *relearning* is able to solve the shortcut issue in the task-id-based method as well.

## C.4 MODEL CALIBRATION

Figures 11, 12, and 13 show the Expected Calibration Error (ECE) plots across three datasets. Without relearning, the model tends to be overconfident, which could be explained by the shortcut features.

Table 4: Time complexity in seconds of our method for CIFAR100-10T

| Library Size | 5k | 10k | 30k | 50k |
|---|---|---|---|---|
| Fetch score | 29.65 | 55.69 | 162.57 | 269.83 |
| Select *library buffer* | 0.05 | 0.06 | 0.10 | 0.13 |
| RL Training | 1604 | 1612 | 1629 | 1624 |

The ECE plots demonstrate that relearning improves model calibration across all three datasets, as indicated by the lower ECE scores.

## C.5 COMPUTATIONAL COST AND RESOURCE

We conduct our experiments using NVIDIA RTX A6000 GPU. The GPU memory consumption depends on the backbone used for training. The experiments should be able to run on a system with at least 8GB of GPU memory when training on Resnet18. The time required to run the experiment varies according to the baseline and the dataset.

Table 4 shows the time required for the important stages of our method. *Fetch score* is the time required to obtain the difficulty score for each sample in the *library*. *Select library buffer* is the time required to select the *library buffer* from the *library*. *RL Training* refers to the time required to update the parameter using *library buffer* for 100 epochs. The results show the overhead of selecting the samples including the score computation is significantly less than the time required to train on *library buffer*. Further, as the *library* size increases, the time to compute the scores increases whereas the time to train on the *replay buffer* remains the same.

## D SOCIAL IMPACT AND LIMITATIONS

In this work, we contribute to improving continual learning, which holds the potential to enable more efficient machine learning tasks including data processing, decision-making, and automation. Our proposed method is generally applicable to any replay-buffer based continual learning strategy. We expect similar limitations as other buffer-based methods because the implementation of relearning strategies may require additional computational resources for data storage and processing, potentially limiting the applicability in resource-constrained environments. The additional storage and computation required are however minimal as we analyzed in the paper. Concerns regarding data privacy and access could restrict the ability to fully utilize these methodologies in sensitive applications.

Our source code is available here.

