# OpenReview forum: "Avoid Being a Shortcut Learner through Library-Based Re-Learning"
_ICLR.cc/2025/Conference — ICLR 2025 Conference Withdrawn Submission_

### Official Review · Reviewer_1y6p · 2024-10-29

**Soundness:** 3
**Presentation:** 3
**Contribution:** 2
**Rating:** 3
**Confidence:** 4

**Summary:**

The author proposes *Relearn*, a relearning method based on *library* to address the shortcut learning problem in CL. *Library* stores the historical information of past tasks. The model is tested with a subset of *library* frequently throughout the training, and relearn the forgotten samples in the library.

**Strengths:**

1. The performance is attractive.
2. The paper is easy to follow.
3. The proposed method makes sense intuitively.

**Weaknesses:**

(Fatal)
1. The author claimed they are trying to solve the shortcut learning problem in the title and introduction. However, there is no evidence in the paper that the shortcut learning problem is addressed with the provided evaluation. In the evaluation, only accuracy is included. To ground the claim soundly, I believe it is necessary to include evaluations of the gradient attention map (or other metrics directly related to shortcut learning, like Shapley value, etc.) in both final results and ablation studies

(Major)
1. Since there are existing works (For example, OnPro in the citation) also aims to solve the shortcut learning problem. Despite the difference in the settings (OnPro is online CL), adapting OnPro to the current setting and making a comparison of accuracies/attention maps should be beneficial to the manuscript.
2. The comparison might be unfair because the Library size (5k) is much larger than the buffer size (200 and 2k) in the evaluation. For fair comparison, having the same information of past tasks (storing the same amount of historical data) is important for fair comparison. Some results of 5k memory size for CIFAR-100-10T are available in [1]

(Minor)
1. There is an inconsistency of Notations in Table 1. (CF100 and CIFAR100) Also, it is suggested to have the same decimal digits in the experimental results.
2. It is suggested that more information about the computation time be included. For example, the proposed method can be compared with other baselines.
3. Stroing a lot of data in the library might be a problem for the CL setting.

(Summary)

The proposed method is interesting and makes sense intuitively. However, the weakness outweighs the strength, and I would like to give a reject.

Reference:

[1] DualPrompt: Complementary Prompting for Rehearsal-free Continual Learning. ECCV 2022

**Questions:**

Please refer to the weakness part.

---

### Official Review · Reviewer_UhPY · 2024-11-02

**Soundness:** 2
**Presentation:** 2
**Contribution:** 2
**Rating:** 5
**Confidence:** 4

**Summary:**

This paper proposes a relearning technique for CL. It select relearning samples considering each sample's learning difficulty. The experimental result table shows improvement.

However, I have some concerns, please see weaknesses.

**Strengths:**

This paper proposes a relearning technique for CL. It select relearning samples considering each sample's learning difficulty. The experimental result table shows improvement.

**Weaknesses:**

1. Please provide more visualization analysis, such as analysis on Task-Recency Bias, Last layer Bias, Feature Drift, etc.

2. In the selection process for the library buffer, Except theoretical analysis, please provide some visualizations to show the different effect brought by selecting different samples to relearning.

3. Figure 3 is not clear. please highlight your novel relearning process in Figure 3.

4. Please compare with more advanced continual learning methods and strategies, such as Mutual Learning/Knowledge Distillation in CL.
[1] Wang, M., Michel, N., Xiao, L. and Yamasaki, T., 2024. Improving Plasticity in Online Continual Learning via Collaborative Learning. In Proceedings of the IEEE/CVF Conference on Computer Vision and Pattern Recognition (pp. 23460-23469).
[2] Michel, N., Wang, M., Xiao, L. and Yamasaki, T., Rethinking Momentum Knowledge Distillation in Online Continual Learning. In Forty-first International Conference on Machine Learning.

5. I am wondering is your relearning process compatible with Mutual Learning/Knowledge Distillation strategies? Please conduct experiments on on the above Refs. [1-2]. By doing so, I believe this paper will be more convincing.

**Questions:**

See Weaknesses

---

### Official Review · Reviewer_PMMD · 2024-11-04

**Soundness:** 2
**Presentation:** 3
**Contribution:** 2
**Rating:** 5
**Confidence:** 3

**Summary:**

This work proposes a strategy for Continual Learning (CL) methods based on rehearsal and the prediction of the Task-id. In this approach, a large "library" of past data expands the replay buffer, allowing the model to test itself on previously learned information. When the model identifies forgotten information or shortcuts (patterns learned for one task that may not generalize to others) the model undergoes "relearning" by training on carefully selected data from the library. The method incorporates a contrastive learning mechanism for the feature extractor and it subsequently updates the classification heads for each task using this sub-sample of the library, referred to as the library buffer.

**Strengths:**

The manuscript is well-written and provides comprehensive analyses, covering several essential aspects such as computational time costs, detailed ablation studies, details on task splitting, and a robust comparison with a diverse set of competitive baselines.
Both Figure 2 and Algorithm 1 make the comprehension of the framework easy, and I appreciate that the methods' performances are averaged over multiple runs, with standard deviations included.

**Weaknesses:**

My main concern about this work is that in a scenario where the literature is increasingly moving towards replay-free approaches (see SLCA[1]), this work appears heavily dependent on retaining *at least* twice the amount of data typically used with a replay buffer.

    [1] Zhang, Gengwei, et al. "Slca: Slow learner with classifier alignment for continual learning on a pre-trained model." Proceedings of the IEEE/CVF International Conference on Computer Vision. 2023.

Additionally, the library, sized at 5k or 10k examples, is nearly as large as the data from an entire task. This goes beyond the typical concept of a replay buffer, as it resembles replaying almost the entire previous task. While I understand that the model is not trained on the whole library, this approach still raises concerns related to privacy and data storage constraints in continual learning settings


**Minor:**

There is a lack of consistency in the notation used for the official names of architectures and methods, and sometimes they are incorrect. For example, "Resnet" should be written as "ResNet," and "Der++" is sometimes referenced this way and other times as "DER++," which is the correct version.

There's a typo in line 538: relay-based should be "replay-based".

**Questions:**

**Questions and points to address**
- To address my main concern regarding fair comparisons, I believe a truly fair evaluation of different baselines’ performance could be achieved by setting both buffer-only and relearning methods to use a total data storage size of $m$. For the relearning methods, this storage could be split into $0.5m$ for the replay buffer and $0.5m$ for the library, (or at least the library buffer) creating a more equitable baseline. This approach would ensure that both methods operate within the same overall memory constraint, avoiding any potential bias from additional memory or data availability for the relearning model.

    Could the authors conduct an experiment where both the buffer-only method (e.g., DER++) and the relearning method (e.g., DER++ + RL) are restricted to the same total storage size? Given time constraints, focusing this setup on a single-method comparison would be sufficient to provide insight.

- How is the threshold $\lambda$ (which determines when to perform relearning based on the quiz mechanism) set for the main Tables
 experiments? The results appear to differ from those presented in Table 3.  I think this value should be explicitly reported for each presented experiment. Additionally, I suggest expanding the explanation of the quiz mechanism in the manuscript, as it currently occupies a relatively small section.

- The buffer and library sizes for the methods listed in Tables 1 and 2 are not very clear, except for the upper part of Table 2. To enhance comprehension of the results, it would be beneficial to include the buffer/library sizes consistently throughout both tables.

- Could the authors clarify which augmentations are applied and specify how many augmented versions of the same input are generated during training? This detail would help in understanding the contrastive learning process more thoroughly.

- I recommend moving the explanation of Figure 1, in particular the section **Empirical evidence for relearning** earlier in the text. By introducing this explanation sooner, the motivations behind the study will become clearer to the reader. A well-placed and timely discussion of the figure can help contextualize the objectives, allowing readers to better progress through the manuscript. This adjustment could enhance the overall flow and comprehension of the work.

 - I have a last question regarding the reported accuracy results for the competing methods. Are these results specifically for methods that do not involve relearning in either Task Incremental Learning (Task-IL) or Class Incremental Learning (Class-IL)? It seems from the text that the results may pertain to Class-IL; however, your method appears to incorporate some form of Task-ID knowledge or estimates.
Clarifying this distinction is crucial, as it will help the reader better understand the validity of the comparisons being made and how these different methods are evaluated against one another. This clarification will enhance the rigour of your analysis and provide valuable context for interpreting the results.

**Details Of Ethics Concerns:**

No Ethics Concerns

---

### Official Review · Reviewer_gKAw · 2024-11-04

**Soundness:** 2
**Presentation:** 3
**Contribution:** 2
**Rating:** 3
**Confidence:** 4

**Summary:**

The paper tackles the problem Class Incremental Learning (CIL), focusing on the shortcut learning problem. To do so, the authors give a theoretical perspective justifying the existence of short learning in Continual Leaning and propose to leverage a large pool of data called library. The library is then used to detect potential shortcut learning issues when learning the current task and allows for relearning. The authors conduct experiments on various relevant CIL datasets and show a considerable improvement compared to state-of-the-art methods.

**Strengths:**

- The information bottleneck perspective of the shortcut learning problem is appreciate
- The performances are compelling
- The core idea is easy to follow
- The code is shared
- The paper is easy to follow

**Weaknesses:**

## Weaknesses
 - l 44. "the replay buffer is fundamentally limited given how it is constructed because the model may be biased when selecting the buffer without accessing the future task data". The replay buffer construction can vary from one paper to the other. No method is specified here, hindering the validity of such argument. In addition, the most common memory building strategy remains the reservoir sampling, which does not rely on the model performances at all, making this argument completely invalid in this context. The authors should be more precise in this statement.
- The entire introduction, while making various claims about the existing strategies, contains only one single reference.
- l52 "Most previous work attributes the inability to retain the previous task data performance as the forgetting issue". This is the actual definition of forgetting.
- l53. The described case of wrong learning or shortcut learning in the case of Continual Learning is not new, see [1, 2]. Such work should be considered for comparison.
- Work on learning representation with mutual information between task have been proposed before and are not mentioned in the paper [3, 4]. The usage of contrastive learning as a mutual information proxy is not new, see also [5].
- l262. What about regularization strategies to mitigate shortcut learning?
- l 311 "Furthermore, just increasing replay-buffer size is not feasible as the computation cost of training on the replay-buffer will increase with the size of replay-buffer." I disagree with this statement. The batch size of data extracted from the buffer will still be the same, so the computation overhead does not depend on the buffer size. However, the space complexity will indeed increase.
- Following on previous point, I do not understand how the authors justify the usage of library while denying large memory size. To me the usage of library is completely similar to using an infinite-size memory buffer and the proposed methods should be compared with larger memory size. Similarly, is computing the difficulty score for each sample in the library computationally intensive?
- The authors claims an improved computation efficiency compared to large memory buffer, but such information is not provided in the paper.
- How do you choose the library size for a given dataset?
- While the paper claims to solve shortcut learning in this context, I do not see muuch evidence of that in the paper.
- Why not compare to [4] ? It is cited in the paper and also addresses shortcut learning. A comparison of various CAM with compared methods, on top of performances, would be required to prove that the proposed method solves shortcut learning.
### citations
[1] Wang, Maorong, Nicolas Michel, Ling Xiao, and Toshihiko Yamasaki. "Improving Plasticity in Online Continual Learning via Collaborative Learning." In _Proceedings of the IEEE/CVF Conference on Computer Vision and Pattern Recognition_, pp. 23460-23469. 2024.

[2] Yujie Wei, Jiaxin Ye, Zhizhong Huang, Junping Zhang, and
Hongming Shan. Online prototype learning for online continual learning. In ICCV, 2023.

[3] Gu, Y., Yang, X., Wei, K., & Deng, C. (2022). Not just selection, but exploration: Online class-incremental continual learning via dual view consistency. In _Proceedings of the IEEE/CVF Conference on Computer Vision and Pattern Recognition_ (pp. 7442-7451).

[4] Guo, Y., Liu, B., & Zhao, D. (2022, June). Online continual learning through mutual information maximization. In _International conference on machine learning_ (pp. 8109-8126). PMLR.

[5] Mai, Z., Li, R., Kim, H., & Sanner, S. (2021). Supervised contrastive replay: Revisiting the nearest class mean classifier in online class-incremental continual learning. In _Proceedings of the IEEE/CVF conference on computer vision and pattern recognition_ (pp. 3589-3599).

**Questions:**

See weaknesses

---

### Note · Authors · 2024-11-22

I have read and agree with the venue's withdrawal policy on behalf of myself and my co-authors.